# BLE-Based Custom Devices for Indoor Positioning in Ambient Assisted Living Systems: Design and Prototyping

**DOI:** 10.3390/s25206499

**Published:** 2025-10-21

**Authors:** David Díaz-Jiménez, José L. López Ruiz, Juan Carlos Cuevas-Martínez, Joaquín Torres-Sospedra, Enrique A. Navarro, Macarena Espinilla Estévez

**Affiliations:** 1Computer Science Department, University of Jaén, 23071 Jaén, Spain; ddjimene@ujaen.es (D.D.-J.); mestevez@ujaen.es (M.E.E.); 2Telecommunication Engineering Department, University of Jaén, 23700 Linares, Spain; jccuevas@ujaen.es; 3Computer Science Department, University of Valencia, 46010 Valencia, Spain; joaquin.torres@uv.es (J.T.-S.); enrique.navarro@uv.es (E.A.N.)

**Keywords:** bluetooth low energy, indoor localization, wearable devices, Zephyr RTOS, fog computing, GATT, reconfigurable systems

## Abstract

**Highlights:**

**What are the main findings?**
Two custom devices were successfully designed and prototyped: a wearable wristband, and a configurable BLE beacon with adjustment of transmission parameters.The laboratory tests confirm stable operation of devices, showing improved RSSI consistency and extended autonomy under different configurations.

**What is the implication of the main finding?**
The development validates the feasibility of tailoring BLE-based devices to overcome the limitations of commercial alternatives, enabling greater control over energy efficiency and signal stability.These advances highlight the potential of the proposed hardware for further refinement and adaptation to real-world experimental setups.

**Abstract:**

This work presents the design and prototyping of two reconfigurable BLE-based devices developed to overcome the limitations of commercial platforms in terms of configurability, data transparency, and energy efficiency. The first is a wearable smart wristband integrating inertial and biometric sensors, while the second is a configurable beacon (ASIA Beacon) able to dynamically adjust key transmission parameters such as channel selection and power level. Both devices were engineered with energy-aware components, OTA update support, and flexible 3D-printed enclosures optimized for residential environments. The firmware, developed under Zephyr RTOS, exposes data through standardized interfaces (GATT, MQTT), facilitating their integration into IoT architectures and research-oriented testbeds. Initial experiments carried out in an anechoic chamber demonstrated improved RSSI stability, extended autonomy (up to 4 months for beacons and 3 weeks for the wristband), and reliable real-time data exchange. These results highlight the feasibility and potential of the proposed devices for future deployment in ambient assisted living environments, while the focus of this work remains on the hardware and software development process and its validation.

## 1. Introduction

The rise of smart environments has driven the development of technological solutions aimed at the localization and tracking of people and assets in various contexts, such as healthcare, industrial, educational, and tourism sectors. Within this framework, wireless technologies have gained prominence, with the Bluetooth Low Energy(BLE) standard standing out for its balance between cost, energy efficiency, and availability in commercial devices [1,2,3,4].

Unlike more specialized technologies such as UWB, RFID, or computer vision, BLE beacons enable the implementation of cost-effective localization systems, particularly in indoor environments where GPS is not operational [5,6,7]. Their operation is based on the periodic broadcasting of advertising packets, whose received signal strength indicator (RSSI) is used by nearby receivers to estimate relative distances or absolute positions through techniques such as trilateration, fingerprinting, or statistical filters [8,9,10].

In the field of indoor localization, BLE technology has established itself as one of the most promising solutions due to its low power consumption, low cost, and broad compatibility with mobile devices. Unlike technologies such as WiFi, BLE can achieve higher accuracy, especially in dense environments where WiFi localization errors can exceed 8 m, compared to less than 2.6 m with BLE in well-designed networks [11].

BLE-based positioning methods have evolved significantly in recent years. The use of beacons in combination with inertial sensors has enabled the development of three-dimensional algorithms such as 3D-LBMS, capable of estimating location with metric-level horizontal accuracy and sub-metric altimetric precision, even in complex indoor environments [12]. On the other hand, lighter systems designed for smartphones leverage fingerprinting maps and built-in sensors to deliver high accuracy without the need for costly infrastructure [13].

A growing trend is the incorporation of machine learning techniques to improve accuracy and robustness against noise in RSSI signals. For instance, the DABIL method employs autoencoders to denoise the signal and enhance three-dimensional localization [14], while decision tree-based techniques and filters such as Kalman have also demonstrated significant reductions in mean error, even in dense Bluetooth environments [15]. Similarly, the use of multiple channels and RSSI signal aggregation has achieved error reductions from 1.5 m to approximately 1 m [16].

The potential of BLE extends beyond traditional positioning. In tourism environments, for example, a robust overlap-tolerant tracking system has been developed to economically and accurately analyze visitor flow in enclosed facilities [17]. Likewise, decentralized approaches that require no prior training have proven practical for rapid and low-cost applications, achieving average errors of 1.5 m [18].

A recent review systematizes the most effective techniques in BLE localization, concluding that sub-meter accuracy is achievable through the combined use of data fusion, filtering, machine learning, and Angle of Arrival (AoA) technology introduced in BLE 5.1 [19]. This review also highlights areas that remain underexplored, such as security and stability under dynamic conditions.

Finally, studies focusing on signal stability indicate that the BLE 5.0 standard specifications offer substantial improvements in RSSI quality and reliability [20], and recent works have identified the potential of BLE for emerging applications beyond traditional beaconing, such as low-latency data transmission and distributed environmental sensing [21].

This principle has also been applied to diverse solutions, including Human Activity Recognition, access management in smart buildings, and patient monitoring in hospitals. However, despite its wide range of applications, commercial beacons present structural limitations that hinder their adaptation to demanding environments [21,22].

One of the main drawbacks lies in the lack of configurability of these devices. Most do not allow the modification of critical parameters such as advertising channel, transmission power, or broadcasting interval. This limitation prevents system optimization in the face of interference, complex architectural geometries, or specific coverage and accuracy requirements. Additionally, physical aspects such as antenna design and device orientation can introduce fluctuations in RSSI, compromising the stability of localization algorithms [23].

Research has shown that the accuracy and energy efficiency of a BLE system directly depend on the proper configuration of its beacons. For instance, it has been observed that channels 37, 38, and 39, used in BLE advertising, exhibit different propagation behaviors due to multipath effects and frequency-specific shadowing. In practical indoor deployments, these distortions are further aggravated by the presence of human bodies [24] and surrounding electronic devices, which act as dynamic obstacles that absorb or reflect radio signals [25]. If the advertising channel is not considered in RSSI analysis, variability is introduced, reducing positioning reliability. Conversely, explicitly identifying the source channel and utilizing all 40 BLE channels can enhance localization accuracy [26].

Transmission power also affects estimation quality. While higher values (e.g., +4 dBm) tend to offer greater accuracy, lower levels (e.g., −20 dBm) can yield acceptable results with significantly reduced energy consumption [27]. Additionally, the advertising interval, which defines how frequently packets are broadcast, plays a crucial role. Shorter intervals improve responsiveness and reduce discovery latency but also accelerate battery drain. Recent theoretical models help optimize these values for latency-constrained applications [28], and adaptive strategies have proven effective in extending device autonomy by up to 200% [29].

In light of these limitations, a custom beacon named BASIA has been developed, enabling adjustment of parameters such as transmission power and advertising channel. It also incorporates an optimized antenna layout to reduce RSSI variability and improve directionality. In parallel, a custom wristband has been developed, capable of scanning beacons and integrating a set of sensors for user monitoring. All collected data are exposed for use in both practical application environments and research scenarios, where full control over transmission conditions is required.

## 2. Motivation and Benchmark Analysis

The development of these devices responds to the need for a localization and monitoring solution that overcomes the limitations of existing systems. The following sections outline the key factors motivating the creation of a proprietary localization system rather than adopting already available solutions.

One of the most relevant factors in opting for a custom development is data management. Existing commercial solutions often rely on third-party platforms, which restrict access to and customization of the collected data. In the proposed system, full control over location and biometric data is maintained, ensuring that only authorized actors defined within the system have access.

### 2.1. Limitations of Commercial Devices

The market offers a wide range of BLE devices designed for indoor localization. However, these devices present certain limitations that impact the system’s stability and accuracy:**RSSI Instability:** Traditional commercial devices transmit advertising packets through the three channels reserved by BLE for this purpose: **37, 38, and 39**. This strategy aims to maximize initial detection by scanners, as each channel operates in a different frequency band to avoid interference from WiFi or other protocols. However, this rotation between channels introduces significant fluctuations in the RSSI, since each channel may experience different propagation conditions and noise levels [30]. This variability complicates accurate distance estimation and negatively affects the stability of RSSI-based localization algorithms—particularly in dynamic environments such as residential settings, where there are multiple reflective surfaces and moving obstacles.Recently, the BLE specification introduced the Extended Advertising mode, available since Bluetooth 5.0, which allows devices to broadcast advertising messages across all available channels in the BLE spectrum, not just channels 37, 38, and 39. This expansion can theoretically improve traffic distribution and reduce congestion, but it also increases RSSI variability by incorporating channels with very different propagation characteristics [24]. Therefore, although Extended Advertising enhances data dissemination capabilities and supports larger payload sizes, its use in received-signal-strength-based localization systems must be carefully evaluated, as it may introduce new sources of signal measurement instability.**Lack of optimization for updating and customization:** Many commercial devices do not support advanced configurations or firmware updates tailored to specific needs. In our system, compatibility with DFU (Device Firmware Update) has been prioritized, allowing two applications to be stored in memory and enabling updates without service interruption. This design allows us to modify device behavior and incorporate new functionalities as required.

These limitations are not exclusive to BLE beacon devices. In fact, although numerous smart wristbands and wearable devices are available on the market for activity tracking and health monitoring, they also exhibit a range of constraints that hinder their effective integration into passive monitoring and indoor localization systems—particularly in residential environments where continuous, low-power, and customizable operation is essential. Among the most relevant limitations are:**High energy consumption and limited autonomy:** Many commercial devices prioritize features such as continuous connectivity, high-resolution displays, or multiple active sensors, which compromises battery life [31]. Passive monitoring applications require extended autonomy and optimized power consumption—requirements that are often not adequately addressed in these products.**Low granularity in data acquisition:** Conventional wristbands often restrict direct access to inertial or biometric sensor data, offering only processed information [31,32,33]. This limitation prevents the development of custom algorithms for activity recognition, posture detection, or physiological event monitoring adapted to the specific needs of the application environment.**Practical impossibility of updating:** Most commercial wristbands do not allow researchers to upload new software directly red [31]. Updates must follow manufacturer-imposed procedures, which are often closed, complex, or completely inaccessible to external teams. This makes it impossible to modify device behavior or add new features suited to the intended context.**Limited compatibility with indoor localization systems:** Since they are not specifically designed for this purpose, commercial wristbands typically lack synchronization and calibration with BLE beacons or anchor nodes, which impacts accuracy in real-time positioning scenarios.

A comparative summary of these limitations in BLE beacons and wearable devices is provided in Table 1.

### 2.2. Evaluation of Commercial BLE Beacons and Wearable Devices: Capabilities and Limitations

Commercial BLE beacons and wearable activity trackers have become increasingly accessible for applications in localization, health monitoring, and ubiquitous computing. These technologies offer notable advantages in terms of hardware reliability, sensor integration, and user-centered design. However, when examined from the perspective of passive monitoring systems in residential environments, particularly those requiring fine-grained control and adaptability, certain limitations become evident.

#### 2.2.1. BLE Beacons

BLE beacons such as the u-blox NINA-B1 (https://www.u-blox.com/en/product/nina-b1-series, accessed on 12 October 2025) module, the iBKS 105 (https://accent-systems.com/es/producto/ibks-105/, accessed on 12 October 2025) by Accent Systems, and the Kontakt.io Nano Tag (https://kontakt.io/products/, accessed on 12 October 2025) have proven effective in a wide range of real-world applications, including asset tracking, personnel monitoring, and indoor navigation in healthcare environments. These devices typically offer configurable parameters like transmission power and advertising interval, as well as support for common BLE formats such as iBeacon and Eddystone. They are well-built and energy-efficient, and their deployment is straightforward using mobile or desktop tools provided by the manufacturers.

The NINA-B1 module from u-blox, for instance, integrates Bluetooth 5.0 capabilities and includes an embedded antenna, offering developers the possibility of firmware customization when working at the embedded level.

The iBKS 105 beacon, manufactured by Accent Systems (Headquarters Castellar del Vallès, Cataluña, Spain), is designed to balance configurability and battery longevity. It allows for parameter tuning via a mobile interface but does not support deeper behavioral changes at the firmware level. Similarly, Kontakt.io offers a family of beacons that integrate cloud-based fleet management and sensor telemetry. Their value lies in centralized monitoring, yet the devices themselves are not intended to be repurposed beyond the vendor’s ecosystem.

While these beacons are robust and well-suited for industrial and commercial deployments, their limited firmware accessibility and lack of support for adaptive or context-aware behaviors reduce their suitability in scenarios that demand high levels of customization. This includes not only research-driven environments, but also personalized monitoring systems in healthcare settings, assisted-living facilities, and smart residential infrastructures. In such applications, the ability to reprogram device behavior, synchronize with other infrastructure elements, or adapt transmission logic in real time is essential for ensuring system reliability and responsiveness.

#### 2.2.2. Consumer Wearables

Devices like the Apple Watch Series 9 (https://www.apple.com/watch, accessed on 12 October 2025), the Xiaomi Smart Band 8 (https://www.mi.com/global/product/xiaomi-smart-band-8/, accessed on 12 October 2025), and the Fitbit Charge 6 (https://www.fitbit.com/global/us/products/trackers/charge6, accessed on 12 October 2025) represent the state of the art in wearable technology. They combine sleek design with sophisticated biometric sensing, including accelerometers, gyroscopes, optical heart rate sensors, and in some models, even blood oxygen and ECG capabilities. For health-conscious consumers, these devices deliver accurate metrics and seamless integration into broader health platforms such as Apple Health or Google Fit.

From a research and system-integration standpoint, however, their strengths come with restrictions. The Apple Watch, for example, offers high-fidelity motion and cardiac sensing, but developers do not have access to raw data streams in real time, nor is it possible to deploy firmware-level modifications or enable custom BLE advertising behavior. Xiaomi’s Smart Band line is attractive for its low cost and long battery life, but data access is limited to what the proprietary app exposes, and BLE communication is often obscured or minimized to save energy. Fitbit’s Charge 6 offers a balance between fitness tracking and data export, but the data it provides through official APIs is aggregated and often delayed, limiting its usefulness in real-time inference or passive monitoring systems.

In all cases, the priority of these platforms lies in privacy, user experience, and device interoperability. While understandable and even commendable from a product design standpoint, these choices restrict their use in experimental or infrastructure-integrated scenarios where low-level control, data transparency, and over-the-air programmability are essential.

#### 2.2.3. Reflections on Suitability for Passive Monitoring Systems

The commercial solutions available today are mature and well-suited to their intended markets. They offer proven stability, refined hardware, and user trust. For systems that need to operate continuously, adapt to changing conditions, and support active firmware development, however, these same strengths become constraints.

Rather than critiquing these products in absolute terms, it is more appropriate to recognize that their design goals differ. They are optimized for end-user satisfaction, health tracking, and managed ecosystems, not for research flexibility or integration into custom BLE-based localization networks. For applications that demand precise control, long-term autonomy, and direct access to sensing and communication parameters, purpose-built or open-hardware platforms remain the more appropriate choice.

## 3. Design and Prototyping of Custom BLE-Based Devices

This section describes the most relevant technical aspects of the developed devices. It presents the design of the printed circuit boards (PCBs) for both the wristband and the beacon devices, as well as the three-dimensional model of their respective enclosures, optimized for functional integration and practical use in real-world environments. Finally, the implemented functionalities of each device are detailed in relation to their role within a proposed localization and monitoring system.

### 3.1. Wristband Device

This section describes the technical design of the wearable device developed for the system. The wristband integrates wireless communication, sensing, and power management into a compact, low-power platform suitable for continuous use in ambient monitoring applications. Its design prioritizes both hardware modularity and ergonomic integration, enabling reliable long-term deployment in real-world environments.

#### 3.1.1. PCB Design of the Wristband Device

The following subsection presents the layout and component selection of the wristband’s PCB. Special attention is given to energy efficiency, sensor integration, and support for firmware updates over-the-air (OTA), ensuring that the device meets the functional requirements of a personalized, adaptive, and transparent monitoring system.

The wristband is a compact system with optimized autonomy and support for over-the-air firmware updates (DFU, Device Firmware Update). Several technical factors influenced the selection of components and the architecture of the PCB, as illustrated in Figure 1.

The wristband integrates two interconnected PCBs: a flexible section and a main rigid board. The flexible board measures 20.65 mm × 17.65 mm, with a two-layer stack-up fully dedicated to signal routing. It hosts 19 components in total, of which 18 are mounted on the top side and only one on the bottom, Table 2 and Table 3. This section is intended to allow the integration of the biometric sensor, adapting to the curvature of the wrist.

The main rigid PCB, by contrast, has dimensions of 30.00 mm × 28.50 mm and follows a four-layer architecture. All layers are dedicated to signal routing, with no internal planes. It integrates 131 components, with 125 placed on the top side and 6 on the bottom. This board accommodates the core system components, including the nRF52833 SoC, the IMU, the PMICs, and the OLED interface. The combination of the rigid and flexible PCBs provides both functional density and mechanical adaptability, which are essential for wearable applications such as health monitoring and ambient assisted living scenarios.

**Main processor:** The initial design considered the nRF52832 (Nordic Semiconductor), a low-power microcontroller with up to 512 kB of Flash and 64 kB of RAM, sufficient for basic BLE applications. However, these specifications proved insufficient for implementing the Device Firmware Update (DFU) mechanism, which requires storing both the active and new firmware versions simultaneously in memory, along with the bootloader. The required space, combined with the additional load from the Bluetooth stack and monitoring functions, exceeded the capacity of the nRF52832.For this reason, the nRF52833 was chosen—a general-purpose multiprotocol SoC based on a 64 MHz Arm Cortex-M4 processor with FPU (Nordic Semiconductor Headquarters Trondheim, Norway), featuring 512 kB of Flash and 128 kB of RAM. This model is qualified to operate within an extended temperature range from −40 °C to 105 °C, and enables OTA (Over-the-Air) updates without compromising system stability. Its BLE radio is compatible with Bluetooth Direction Finding and supports a transmission power of up to +8 dBm, ideal for improving indoor coverage.Additionally, it includes advanced interfaces such as USB 2.0 Full-Speed (12 Mbps), NFC-A, 32 MHz SPI, ADC, I^2^S, PDM, UART, PWM, and a wide set of digital buses. Its broad supply range (1.7 V to 5.5 V) allows operation with both rechargeable batteries and USB sources, while its automated power management system with dual-stage LDOs and an integrated DC–DC converter ensures efficient power consumption even in demanding applications. These characteristics make the *nRF52833* an optimal choice for advanced wearable devices such as the ASIA wristband.**User interface and energy efficiency:** A 1″ SPI OLED display (WINSTAR Display (Taichung, Taiwan)) was integrated, offering lower power consumption compared to traditional LCDs. Thanks to OLED’s ability to turn off individual pixels, power usage can be optimized by displaying only essential information and avoiding constant backlighting. Furthermore, the display is updated efficiently to reduce the processor and SPI bus load.**Sensor stage:** An inertial unit LSM6DSM (STMicroelectronics (Geneva, Switzerland)) is included, featuring a 3-axis accelerometer and gyroscope for detecting user motion and posture. In addition, a biometric sensor *PAH8151HU* (Pixart (Hsinchu, Taiwan)) allows heart rate and oxygen saturation measurement through photoplethysmography (PPG), with an optimized sampling algorithm to minimize consumption in inactive mode.**Power stage:** A linear battery charger MP2662 (Monolithic Power Systems (Kirkland, WA, USA)) with integrated power path is used, enabling uninterrupted system operation during charging. A fuel gauge LC709204F (ON Semiconductor (Scottsdale, AZ, USA)) provides accurate battery level monitoring, while a buck-boost regulator ISL9122AIINZ(Renesas (Tokyo, Japan)) stabilizes the output at 3.3 V for powering critical modules.**PCB design and thermal dissipation:** The PCB layout was optimized to minimize interference and ensure efficient power distribution. Dedicated ground planes were implemented to enhance RF signal stability and reduce susceptibility to electromagnetic noise. Thermal vias were added under high-consumption components, such as the voltage regulator, to facilitate heat dissipation.

#### 3.1.2. 3D Enclosure Design

In addition to the electronic architecture, the physical form factor of the wristband was a key design element. The 3D design of the wristband device followed an iterative process focused on achieving optimal integration between electronics, usability, and user comfort in passive monitoring contexts. Long-term wearability imposed strict requirements on ergonomics, sensor stability, and display accessibility.


**Initial Design Options**


During the conceptualization phase, two main configurations were considered based on the orientation of the integrated OLED display:**Vertical display layout:** This format aligned the screen parallel to the forearm axis. It allowed more natural reading when the arm was relaxed or at rest, and offered a sleeker, more discreet appearance.**Horizontal display layout:** The screen orientation perpendicular to the forearm facilitated direct viewing when rotating the wrist and allowed buttons or indicators to be placed on the sides of the central module.

After several iterations and usability tests, the horizontal design was chosen, as it offered better integration with the wrist’s natural shape and allowed for a suitable balance between screen area, device size, and sensor accessibility, as shown in Figure 2. This decision also reduced the necessary curvature of the electronic module, improving sealing and simplifying manufacturing.

The final version features interchangeable silicone straps and a moisture-resistant technical housing, enabling continuous use in domestic, clinical, or light activity settings. User comfort during prolonged wear was prioritized, ensuring correct positioning of sensors such as the photoplethysmograph (PPG) and the IMU to achieve stable readings without requiring active user interaction.

The mechanical enclosure of the wristband has been designed with compactness and wearability in mind. According to the technical drawing, shown in Figure 3, the device exhibits the following dimensions:**Overall height**: 93.54 mm**Width**: 68.4 mm**Body thickness**: 24.37 mm**Maximum thickness with central protrusion**: 27.08 mm**Display area**: 52.28 mm × 32.44 mm

### 3.2. Beacon Device

This section presents the fixed BLE beacon developed for the system. Although it shares many architectural elements and components with the wristband device, the beacon has been specifically designed for continuous operation, minimal maintenance, and straightforward deployment in indoor environments. As a static broadcasting node, its design emphasizes robustness, modularity, and unobtrusive integration into real-world settings.

#### 3.2.1. PCB Design of the Beacon Device

The beacon device functions as a fixed node that transmits BLE advertising packets. Since it shares several components with the wristband, its PCB was designed to simplify production and maintenance, as shown in Figure 4. The PCB of the ASIA Beacon prototype (Table 4) was designed with compactness and manufacturability in mind. The board measures 49.60 mm in width and 81.90 mm in length, making it suitable for integration into medium-sized stationary enclosures. The design adopts a four-layer stack-up, where all layers are dedicated to signal routing; no internal planes were defined at this stage, as the prototype emphasizes flexibility in rerouting and testing different sensor configurations.

A total of 145 electronic components were integrated. The majority are mounted on the top side of the board (124 components), including the nRF52833 SoC, power management ICs, sensors, and connectors. The bottom side hosts 21 components, mainly passives and auxiliary circuitry, in order to optimize surface utilization while maintaining assembly feasibility. This distribution reflects a design trade-off between minimizing footprint and ensuring accessibility for debugging and testing during the prototyping phase.

**Main processor:** The nRF52833 was retained to ensure full compatibility with the wristband software and to facilitate unified development and code maintenance across devices.**User interface:** The system integrates six white LEDs as visual status indicators, along with an RGB status LED controlled by the LP5009 driver (Texas Instruments (Dallas, TX, USA)), providing clear feedback on system activity and behavior.**Power stage:** The device uses the same ISL9122AIINZ regulator and LC709204F fuel gauge as the wristband. However, a switching charger, the MP2723 (MPS), was chosen for this application to enable faster and more efficient charging, which is crucial for a continuously operating device.**Mechanical integration and mounting:** The PCB was designed to fit within a discrete enclosure, allowing installation on walls, ceilings, or furniture without disrupting the aesthetic of the environment. The device eliminates unnecessary user-facing elements to reduce interaction and support a low-maintenance profile.

#### 3.2.2. 3D Enclosure Design

Alongside the electronic design, the 3D housing of the beacon device underwent an iterative development process, aiming to achieve optimal integration between internal components and the external appearance. The design considered factors such as device discretion in indoor environments, ease of installation, and ergonomic handling during setup or repositioning.


**Initial Design Options**


During the conceptual phase, two main approaches for the device enclosure were evaluated:**Vertical enclosure with soft contours:** A vertically oriented housing with rounded edges, intended for seamless aesthetic integration into domestic spaces.**Horizontal watch-like enclosure:** A more compact and stylized design, horizontally oriented to facilitate installation on a wider variety of surfaces.

After evaluating both options, the vertical design was selected for its improved discretion and greater installation versatility, as illustrated in Figure 5. This version supports two main mounting configurations:**Wall-mounted:** Secured with mechanical anchors in an elevated position, maximizing the coverage and detection range.**Surface placement:** Thanks to its stable base, the device can be positioned on furniture such as tables or shelves without the need for permanent fixtures.

The beacon unit’s principal dimensions, illustrated in Figure 6, are:**Height**: 94.6 mm**Width**: 56.7 mm**Depth (base to button surface)**: 49.52 mm

## 4. System Architecture Based on Custom BLE Devices

The proposed system for indoor localization is built upon a distributed architecture composed entirely of custom-designed BLE devices. Specifically, the ASIA Beacon and wristband were developed to operate autonomously in indoor environments, offering high configurability, energy efficiency, and modularity.

Unlike commercial systems, whose internal logic is often encapsulated and non-modifiable, the devices introduced in this work provide full control over their behavior. This includes BLE advertising parameters, transmission power, channel selection, and exposure of sensor data through standardized interfaces such as GATT (Generic Attribute Profile). These features make the system suitable for both practical deployments and experimental setups requiring fine-grained instrumentation.

All devices operate under a real-time logic governed by the Zephyr RTOS, which manages periodic tasks, hardware interrupts, and work queues. This allows for precise timing, low-latency event handling, and scalable modular integration of sensors and user interfaces. Figure 7 illustrates the high-level system architecture based on these devices. The decision to adopt Zephyr RTOS instead of alternatives such as FreeRTOS is mainly based on its open and less restrictive licensing model, its rapidly growing community, and the strong support provided by the semiconductor industry. Zephyr is released under the Apache 2.0 license, which offers greater flexibility for both commercial and academic projects by reducing legal constraints and simplifying integration with proprietary components. Moreover, the project benefits from a large and expanding ecosystem backed by the Linux Foundation, where contributions from semiconductor companies, OEMs, and independent developers ensure continuous improvement, peer-reviewed quality, and a rich set of resources. A critical factor is the explicit support from Nordic Semiconductor, the manufacturer of the target SoCs, which guarantees long-term compatibility, optimized drivers, and vendor-maintained features that accelerate development and reduce risk.

### 4.1. Functional Logic of Devices

This section presents the functional logic embedded in each device. The design is based on a modular and event-driven architecture implemented using Zephyr RTOS, which enables reliable timing, efficient resource management, and deterministic behavior across all hardware subsystems. Each device operates independently, yet in synchrony with the broader system, fulfilling complementary roles: fixed signal broadcasting in the case of the beacon, and mobile contextual sensing and data exposure in the case of the wristband.

#### 4.1.1. Beacon

The beacon is a low-power autonomous node designed to periodically broadcast contextual information through BLE advertising. Its logic is optimized for residential and indoor environments, prioritizing energy efficiency, signal stability, and configurability.

Its main functionalities are:**Custom BLE advertising packets:** The beacon transmits encoded data via the *manufacturer data* field, using Nordic Semiconductor’s identifier. The payload includes transmission power, active advertising channel, battery level, supply voltage, internal temperature, and device status flags (e.g., active charging, low-power mode). These packets are broadcast periodically without requiring a persistent connection.**Dynamic BLE channel management:** The device can switch between BLE advertising channels (37, 38, 39), allowing evaluation of signal quality and minimizing interference. This behavior can be manually triggered via a physical button.**Adaptive transmission power control:** The beacon can cycle through different BLE transmission power levels (from −20dBm to +8dBm) via manual press button. This feature enables optimization of range, power consumption, or localization resolution.**Energy and environmental monitoring:** Integrated sensors measure battery Relative State Of Charge (RSOC), voltage, and internal temperature. Charging state (active, full, or fault) is also tracked. All parameters are updated regularly and included in advertising packets to provide real-time telemetry.**User interface and local interaction:** A physical button allows the user to change the BLE channel and transmission power. RGB and white LEDs provide visual feedback, such as battery level or device state.**Timers and scheduled tasks:** Device logic relies on Zephyr timers to control watchdog refresh, sensor polling, and advertisement updates. This ensures autonomous, low-maintenance operation.

#### 4.1.2. Wristband

The wristband functions as a mobile node within the system. It collects both environmental and internal data, exposing this information through a custom GATT service. It integrates BLE scanning capabilities with biometric and inertial sensing.

Its main functionalities are:**BLE environment scanning:** The wristband continuously scans for BLE advertising packets from beacons. It extracts data such as RSSI, advertising channel, and transmission power, which are used for proximity estimation and environmental awareness.**GATT-based data exposure:** All collected data are made available through a custom GATT service, allowing access from edge nodes or mobile apps without requiring persistent pairing, facilitating interoperability.**Internal status monitoring:** The wristband tracks its own battery level, supply voltage, internal temperature, and charging state. These data are periodically updated and available via GATT.**Physiological and motion data acquisition:** An integrated 6-axis IMU (accelerometer and gyroscope) and a PPG sensor allow the wristband to capture movement patterns, heart rate, and blood oxygen saturation (SpO_2_). Although useful for activity monitoring, in this system these data contribute to context-awareness and localization refinement.**Compatibility with distributed architectures:** The wristband connects directly to an edge node (e.g., Raspberry Pi), which gathers GATT data and concurrently scans for beacon advertisements. This enables central data collection and local processing.**Adaptable firmware design:** The firmware can be customized to adjust sampling rates, scanning behavior, and sensor activation policies, adapting to different indoor localization scenarios.**On-device display:** An integrated OLED screen displays key user data such as battery level, step count, and current time.**Minimal user interaction:** A single physical button activates temporary display of parameters. This minimal interface avoids distractions and supports passive usage.

### 4.2. Data Collection and Processing Architecture

The system follows a decentralized architecture, where a fog computing node, implemented using a Raspberry Pi, serves as the intermediary between custom BLE peripherals and the backend infrastructure. This node is responsible for maintaining a BLE link with the wearable device in order to retrieve contextual data via a GATT interface (e.g., RSSI values, battery level, heart rate). Simultaneously, it passively scans the environment to detect advertising packets emitted by beacons.

All the information acquired is organized and transmitted using the MQTT protocol. This enables efficient local processing, buffering, and asynchronous delivery to a centralized server where the data can be stored, visualized, or further analyzed. This architecture also supports the deployment of lightweight localization algorithms directly on the edge node, improving system responsiveness and autonomy.

Within this framework, beacons play a central role as stationary transmitters that periodically broadcast relevant telemetry data. Each beacon encodes information into a compact Manufacturer Specific Data (MSD) field, structured to reflect key operational and environmental parameters. Table 5 outlines the layout of the MSD payload.

The beacon’s BLE behavior can be adapted at runtime. Among the configurable parameters:**Transmit power adaptation:** adjustable between multiple levels, ranging from −20 dBm to +8 dBm, enabling control over coverage and energy usage.**Channel rotation:** automatic or manual cycling through advertising channels 37, 38, and 39 to reduce interference or enable signal quality comparison.

Interaction is facilitated through onboard physical buttons, which trigger hardware interrupts linked to dedicated routines. These routines are executed asynchronously using Zephyr’s kernel timer and work queue infrastructure, supporting actions such as adjusting transmission power, switching advertising channels, or toggling LED feedback.

The full BLE advertising packet structure is presented in Table 6, highlighting the combination of standard fields with the custom MSD segment.

The wristband exposes a custom GATT service that enables external systems to access internal state and sensing data in real time. Designed for interoperability with edge devices and mobile platforms, the service supports both polling via read operations and asynchronous updates via GATT notifications.

All characteristics are grouped under a primary service identified by UUID 0xA100. This includes metrics related to power management, inertial movement, cardiovascular signals, step count, and BLE-based proximity to beacons.

Table 7 summarizes the set of available characteristics, including their UUIDs, data formats, units, and access capabilities.

The Beacon RSSI List characteristic dynamically reports proximity estimates based on BLE scans. Each element in the list contains:mac_addr(6 bytes): Beacon’s BLE MAC address.timestamp (uint32_t): Time of reception in milliseconds.rssi (int8_t): Signal strength in dBm.

The PPG Raw Data characteristic provides access to unfiltered photoplethysmographic signal samples acquired from the optical sensor. Each notification transmits an array of signed 16-bit integer values corresponding to raw ADC readings. These data can be used for downstream signal processing tasks such as heart rate variability (HRV), SpO_2_ estimation, or pulse waveform analysis.

All characteristics are capable of emitting notifications, allowing subscribed clients to receive timely updates without the need for continuous polling. This design optimizes bandwidth efficiency and supports real-time monitoring scenarios.

## 5. Security Considerations

Security has been treated as a transversal aspect in the design of the proposed architecture, from the communication of the wearable devices to the management of data in the backend. At the lower level, BLE connections implement the standard security mechanisms to prevent unauthorized pairing and to guarantee confidentiality of characteristic exchanges. This ensures that the sensor data exposed by the custom GATT service of the wristband is protected at the link layer.

Inside the home, the fog node plays a critical role as a trusted gateway. It aggregates data from the wearable and beacon through GATT and advertising channels, and transmits these streams to the other services using the EMQX broker. All communications between the fog node and the backend are encrypted with TLS 1.3, providing end-to-end confidentiality, integrity, and forward secrecy during data transmission. This design ensures that even if the underlying network is untrusted, the contents of the data streams cannot be intercepted or modified.

At the backend, access to the MongoDB database is also secured through TLS 1.3. The database is protected behind a hardened stream proxy that enforces strict cipher suites, short connection timeouts, and certificate validation. Beyond transport security, encryption is applied at the disk level to protect stored data against physical compromise. We are also evaluating database-native encryption strategies, with the goal of providing an additional layer of fine-grained protection for collections containing sensitive biometric or localization information.

Perimeter defense is provided by a reverse proxy that terminates TLS connections using modern algorithms such as ChaCha20–Poly1305 and AES–GCM, and enforces HTTP/2 and HTTP/3 with HSTS policies. Rate limiting, connection quotas, and strict header validation are applied to mitigate denial-of-service attempts and to reduce the risk of protocol exploitation. These policies are complemented by the removal of unnecessary server tokens and the denial of access to hidden or administrative paths.

Operational security measures also include detailed monitoring and logging. All access attempts, proxy errors, and upstream responses are logged with extended information such as request times, user agents, and IP addresses. These logs are used not only for anomaly detection in real time, but also for forensic analysis in the event of suspected incidents.

A key principle guiding the system is data sovereignty. All collected information is stored exclusively in servers managed by the research team, avoiding dependence on external cloud providers. This approach minimizes the exposure of sensitive data to third parties and ensures that processing remains within a controlled infrastructure. The combination of link-layer protection in BLE, end-to-end encryption with TLS 1.3 for fog-to-server communication, hardened database access, and strong perimeter policies result in a multilayer security strategy adapted to the needs of ambient assisted living environments.

To reduce re-identification risk while preserving analytical value, all records stored and transported by the system use pseudonymous identifiers. Each subject and device is assigned a unique, non-semantic ID (e.g., UUIDv4) that appears in GATT/MQTT payloads, topics, and database documents; no directly identifying attributes (name, email, address) are included in telemetry streams. The real-world linkage (the correspondence between the pseudonymous ID and personal identifiers) is kept in a separate, access-controlled mapping table, stored under a distinct schema and protected by encryption at rest and in transit. Access to this table is strictly limited to authorized operators under least-privilege roles and audited procedures. For operational logs and analytics exports, identifiers are further hardened via keyed hashing (HMAC) or irreversible tokenization, ensuring that accidental exposure of secondary datasets does not reveal identities. This design confines personal data to a minimal, well-guarded boundary while allowing the rest of the platform to operate with pseudonymized data only, aligning with data minimization and purpose limitation principles.

### 5.1. Advantages of the Proposed Architecture

This BLE-based system architecture, grounded on custom hardware and open interfaces, provides the following advantages:**Full control and transparency:** Device behavior can be modified at the firmware level, allowing fine-tuning and complete experimental instrumentation.**Energy efficiency:** Optimized sleep schedules and low-power components enable long-term operation with minimal maintenance.**Scalability and modularity:** Devices can be deployed incrementally or reconfigured in-place without affecting system stability.**Integration flexibility:** GATT and MQTT interfaces enable seamless integration with edge computing platforms and cloud services.**High-resolution contextual data:** The system provides synchronized, multi-modal data suitable for advanced indoor localization models and behavior inference in future applications.

### 5.2. Limitations and Possible Drawbacks

Despite the advantages demonstrated by the proposed BLE-based architecture, several challenges and potential drawbacks must be acknowledged:**RSSI stability:** Although selective channel strategies reduced variability, maintaining stable signal behavior in dynamic architectural layouts remains difficult, particularly in residential environments with reflective surfaces and moving obstacles.**Energy autonomy:** Current autonomy levels range from 3–4 months for beacons to 2–3 weeks for the wristband. Achieving months-long operation under dense deployments with short advertising intervals continues to be an open challenge.**Data security and robustness:** While GATT and MQTT interfaces provide flexibility, ensuring secure exchange of sensitive data and resilience against wireless interference requires further refinement.**Interoperability:** Integration with heterogeneous commercial IoT devices is not always seamless, demanding additional efforts to guarantee compatibility with diverse architectures.**Device size and ergonomics:** While the custom wristband and beacon provide full configurability, their current form factor is bulkier compared to miniaturized commercial devices. This may affect user comfort and acceptance in long-term deployments.**Maturity of the system:** Both software and hardware are currently under refinement for potential commercialization. Long-term testing in inhabited smart environments is still required to validate performance and usability under realistic conditions.

These limitations highlight the need for continuous improvements in stability, energy efficiency, and interoperability to ensure large-scale, long-term adoption of the system in ambient assisted living scenarios.

## 6. Experimentation and Results

The system is currently undergoing functional testing at the SmartLab of the University of Jaén, an experimental smart environment that simulates realistic conditions in intelligent home scenarios. In this setting, key performance indicators are being evaluated, including RSSI stability, detection latency, energy autonomy, robustness against interference, and integration capability with edge nodes and third-party sensors.

Table 8 summarizes the main software tools and versions used during the experimentation stages.

### 6.1. Advertising Rate Evaluation Using nRF52840 Dongle

To evaluate the effectiveness of the advertising protocol under different timing configurations, a series of measurements was collected using the nRF52840 Dongle (PCA10059) by Nordic Semiconductor. The beacon was configured to broadcast advertising packets at nominal intervals of 50 ms, 100 ms, and 500 ms. Actual measured intervals were 55.00 ms, 105.00 ms, and 505.00 ms, respectively, due to stack-level timing granularity and scheduling overhead.

Each test was executed over a duration of 60 s, with the beacon set to advertise on all three BLE advertising channels (37, 38, and 39), producing up to three packets per advertising event. A second nRF52840 Dongle, operating in continuous scanning mode, was used to capture the transmitted packets.

As presented in Table 9, the measured packet emission rates were approximately 60.0, 30.0, and 6.0 packets per second for the 50 ms, 100 ms, and 500 ms configurations, respectively. The BLE stack reported a consistent advertising event duration of 3.39 ms across all cases. This translates into radio occupancy ratios (active time per interval) of 6.78 %, 3.39 %, and 0.68 % for the 50 ms, 100 ms, and 500 ms intervals, respectively.

Packet reception results, summarized in Table 10, showed a capture rate of 95 % for the 50 ms configuration, and 100 % for both 100 ms and 500 ms. These variations reflect the influence of scan window alignment and processing time within the receiving stack. Although shorter intervals slightly increase the probability of missed packets, they also deliver much higher temporal resolution.

In particular, the 50 ms configuration yielded over 3100 captured packets per minute, enabling high-frequency telemetry well suited for fine-grained indoor localization and real-time sensor monitoring, provided the receiving hardware supports sufficient throughput.

### 6.2. Estimated Power Consumption Under BLE Advertising Scenarios

To estimate the power profile of the system, simulations were conducted using the Online Power Profiler for Bluetooth LE provided by Nordic Semiconductor. Although this tool is designed for the nRF52832 (https://www.nordicsemi.com/Products/nRF52832, accessed on 5 July 2025) chip, it offers a reliable approximation of the expected behavior for our hardware platform, which is based on the nRF52833 (https://www.nordicsemi.com/Products/nRF52833, accessed on 5 July 2025) SoC. Both chips share the same core architecture (ARM Cortex-M4F) and similar radio and power management characteristics, making the estimation valid for comparative and design-level purposes.

The power profiling was configured under the following conditions: a supply voltage of 3.3 V, activation of the DC–DC regulator, use of an external crystal for the low-frequency clock, and a radio transmission power of +4 dBm. The BLE mode was set to non-connectable advertising, with a fixed advertising packet size of 18 bytes transmitted at an on-air data rate of 1 Mbps.

Three different advertising intervals were evaluated: 50 ms, 100 ms, and 500 ms. These represent typical configurations for latency-sensitive, balanced, and energy-optimized operation modes, respectively. The results are summarized in Table 11.

As shown in Table 11, the average current consumption drops significantly as the advertising interval increases. At 50 ms, the system consumes an estimated 188 μA, while extending the interval to 500 ms reduces the average current to just 22 μA. These figures highlight the strong impact of advertising frequency on energy efficiency.

It is important to note that these results are theoretical and do not capture the full behavior of our system, which includes custom firmware, beacon identification logic, and data field updates. However, the estimations provide a realistic lower bound and serve as a reference point for selecting trade-offs between update rate and autonomy during system configuration.

### 6.3. Experimental Anchor Configurations with Variable Transmission Power

To complement the theoretical estimations, preliminary experiments were conducted using multiple anchor configurations under different radio transmission power levels. The advertising interval was fixed at 300 ms, which serves as the initial reference value for system evaluation. This interval was selected as a compromise between responsiveness and energy efficiency, ensuring sufficient update frequency while preserving autonomy.

Three transmission power levels were tested: +8 dBm, 0 dBm, and −20 dBm. These values were chosen to explore the performance envelope of the hardware, ranging from maximum coverage to highly constrained low-power operation. The resulting configurations enable the analysis of trade-offs in terms of communication range, energy consumption, and potential interference resilience.

The anchors were physically deployed and kept in a static position, while their battery levels were regularly monitored through the status reports they broadcast. Over an observation period of 3–4 months, corresponding to a semi-mature stage of the firmware, significant differences were detected among the tested configurations, with about a 15% deviation in battery level across anchors. This suggests that, under the current software implementation, the chosen reference parameters can impose a prohibitive energy burden.

Nevertheless, several improvements could be introduced to further reduce power consumption. For instance, it remains an open question whether all anchors require a strict 300 ms advertising interval. It is conceivable that certain anchors could operate at longer intervals without compromising the system’s overall performance, thereby extending battery lifetime. Such adaptive configurations could become a valuable design strategy in future iterations, balancing system responsiveness and energy autonomy depending on deployment context.

### 6.4. Comparison with Commercial Beacons

To highlight the advantages of the proposed ASIA Beacon, we provide a comparative analysis against a widely used commercial reference, the iBKS 105 from Accent Systems. Table 12 summarizes the main technical aspects and functional differences.

As observed, while the iBKS 105 offers robustness and long autonomy for conventional beaconing applications, it lacks the flexibility required for adaptive indoor positioning and experimental research. In contrast, the ASIA Beacon provides full firmware control, telemetry, and runtime configurability, enabling its integration into scalable ambient assisted living scenarios. The comparatively higher costs of the ASIA Beacon at this stage are mainly due to the expensive production process: custom 3D-printed enclosures amount to around EUR 25 per unit, whereas with injection molding, the same casing could be manufactured for less than EUR 1, according to preliminary estimates. Likewise, the electronic components are currently procured in very small batches, and the assembly process is costly, but both factors are expected to decrease drastically in a medium-scale production scheme.

### 6.5. RSSI Stability Comparison

A preliminary experiment was conducted to compare the signal stability of the proposed ASIA Beacon against a commercial iBKS 105 device. Both beacons were placed under controlled conditions in an anechoic chamber (Figure 8), and the RSSI was recorded using a Samsung smartphone as receiver. The tests were carried out in the Anechoic Chamber of the Radiofrequency and Antennas Laboratory at the University of Valencia, one of the largest and most sophisticated facilities of its kind in Spain. This 5.5 × 8.5 × 5.5 m installation enables high-precision measurements in interference-free environments, which are essential to validate the performance of the indoor positioning system. The walls are covered with absorbent panels, which consist of pyramids of absorbent material that are 50 cm high in the central area of the walls. These panels reflect between −45 and −50 dB of the incident radiation at the devices’ frequencies. The Orbit/FR positioners (https://www.mvg-world.com/es/orbit-fr (accessed on 12 October 2025)) also allow the devices to be precisely positioned inside the chamber with the AL-48062 Positioning Controller. This facility allows the characterization of antennas and wireless sensing devices from 300 MHz up to 110 GHz, supporting both far-field spherical and near-field planar measurements, thus providing an ideal environment for controlled validation. Moreover, the controlled conditions of the chamber make it possible to analyze multipath propagation effects and frequency-dependent attenuation in a reproducible manner—factors that are critical for evaluating BLE-based positioning systems, where even small variations in RSSI can compromise localization stability. The use of this facility, therefore, provides a reliable baseline for comparing the behavior of commercial and custom-developed beacons. In this setup, the ASIA Beacon was configured to transmit exclusively on a single advertising channel (ch. 37), whereas the iBKS 105 operated in its default mode, cycling across all three advertising channels (ch. 37–39). Table 13 summarizes one representative dataset of the ongoing tests.

In this setup, the ASIA Beacon was configured to transmit exclusively on a single advertising channel (ch. 37), whereas the iBKS 105 operated in its default mode, cycling across all three advertising channels (ch. 37–39). Table 13 summarizes one representative dataset of the ongoing tests.

As observed, the iBKS 105 exhibited a significantly weaker average signal (approximately 15 dB lower) and higher variability compared to the ASIA Beacon. The boxplot and histogram analysis (Figure 9) confirm that the proposed beacon achieves a more concentrated distribution of RSSI values, thus providing greater stability for localization purposes.

For the data shown in Table 13, each mean RSSI value was obtained from a set of measurements taken as part of the beacon characterization procedure. The antenna response was evaluated over the plane, ranging from 0° to 360° in increments of 30°. At each angle, approximately 100 RSSI samples were collected, and multiple rounds of measurements were performed to ensure repeatability. The dataset presented in Table 13 corresponds to one representative measurement round, which consistently maintained the same behavior across angles. It is important to highlight that this dataset corresponds to a small sample of a broader set of experiments performed in the anechoic chamber, including different beacon orientations and transmission power levels. These preliminary results reinforce the potential of the ASIA Beacon to deliver more stable RSSI measurements under controlled conditions, which will be further validated in an extended test under different conditions.

### 6.6. Conclusions of Preliminary Experimental Results

Although further experimentation and long-term measurements are still required to complete the signal characterization, initial findings are encouraging:A **reduction in RSSI variability** observed when employing selective channel strategies.An estimated battery life of 3 to 4 months for ASIA beacons operating continuously.A battery lifetime of approximately 2 to 3 weeks for the wristband under continuous monitoring usage.Confirmed compatibility with distributed IoT architectures, leveraging GATT for data exposure and MQTT for reliable transmission to edge nodes or external servers.

The proposed devices have shown solid performance in terms of system stability, minimal maintenance requirements, and flexible deployment capabilities. These early results establish a reliable foundation for future field validation in real-world, inhabited smart environments.

However, several potential drawbacks and open issues must be considered:**Need for large-scale testing:** Current evaluations have been carried out under controlled conditions. Real deployments in inhabited environments are essential to assess system behavior under diverse architectural layouts, user mobility patterns, and interference from coexisting wireless systems.**Impact of different surfaces and materials:** RSSI fluctuations strongly depend on the presence of reflective or absorbent surfaces (e.g., concrete walls, metallic furniture, glass partitions). Understanding these effects requires extensive experimental campaigns in heterogeneous residential and clinical environments.**Advertising trade-off with single-channel operation:** When restricting transmission to a single advertising channel, the number of broadcast packets is effectively reduced to one third. Therefore, in order to maintain high detectability, it becomes necessary to reduce the advertising interval so that a sufficient number of packets are emitted over time.**Energy efficiency improvements:** Although autonomy results are promising, further optimization is needed to reach the efficiency levels of industrial solutions. Adaptive duty-cycling, intelligent advertising policies, or energy harvesting mechanisms could help extend lifetime in dense networks.**System maturity:** Hardware and firmware are still under refinement for potential commercialization. Additional effort is required to ensure robustness.

## 7. Conclusions

The architecture and embedded devices developed in this work constitute an integrated solution for indoor localization and monitoring, with a particular emphasis on energy efficiency, customization capability, and functional transparency.

Unlike existing commercial solutions, the custom-designed beacons and wristbands provide fine-grained control over critical parameters such as BLE channel selection, transmission power, advertising frequency, and access to raw sensor data. This openness has made it possible not only to improve system accuracy, but also to adapt its behavior dynamically to the environmental conditions and the specific needs of each application.

From a scalability perspective, the proposed system shows strong potential for deployments ranging from small experimental testbeds to large-scale assisted-living infrastructures. The combination of OTA firmware updates, standardized interfaces (GATT/MQTT), and energy-efficient design could enable incremental integration into existing IoT platforms with minimal maintenance overhead. These features suggest that the system has the capacity to achieve long-term viability in real-world smart environments.

Nevertheless, several challenges remain for large-scale and long-term deployment. Maintaining signal stability in dynamic architectural layouts, ensuring secure data exchange, and achieving months-long autonomy in dense deployments are open issues that require continuous refinement. Moreover, robustness against interference and interoperability with heterogeneous commercial devices are critical aspects to be addressed in future iterations.

At present, the devices are in a refinement stage, undergoing evaluation for potential commercialization. Both software and hardware components are being polished for upcoming revisions, and extensive testing needs to be carried out across different environments (e.g., residential installations, assisted-living facilities, and experimental laboratories) to assess system reliability and usability in realistic conditions.

Future directions will focus on extending field trials in inhabited smart environments, exploring advanced localization algorithms (e.g., fusing inertial and mmWave data with RSSI-based positioning), and integrating intelligent services for passive activity recognition. Additionally, efforts will be directed toward improving miniaturization, optimizing energy consumption, and ensuring compliance with industrial standards to facilitate eventual market adoption.

These results are especially relevant for applications such as teleassistance, elderly monitoring, and smart homes, where reliable behavior and extended autonomy are essential. Future work will focus on real-world deployment to evaluate long-term performance, usability, and integration with intelligent services for passive activity recognition.

## Figures and Tables

**Figure 1 sensors-25-06499-f001:**
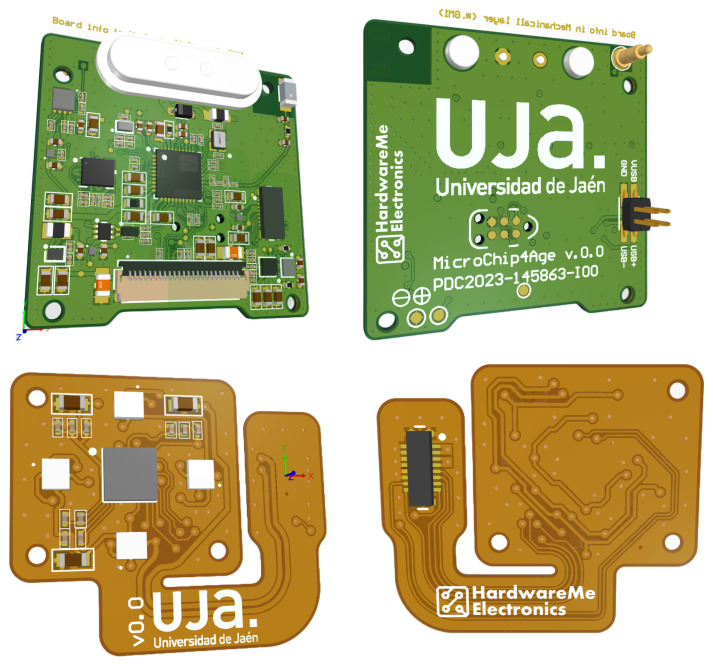
Main PCB and flexible PCB.

**Figure 2 sensors-25-06499-f002:**
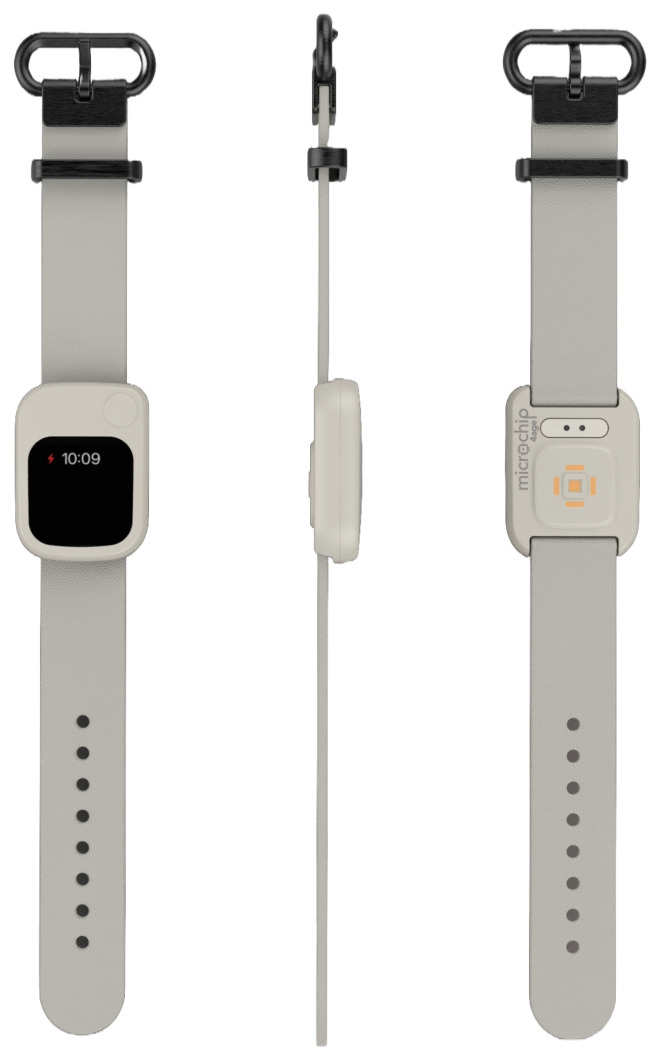
Wristband housing.

**Figure 3 sensors-25-06499-f003:**
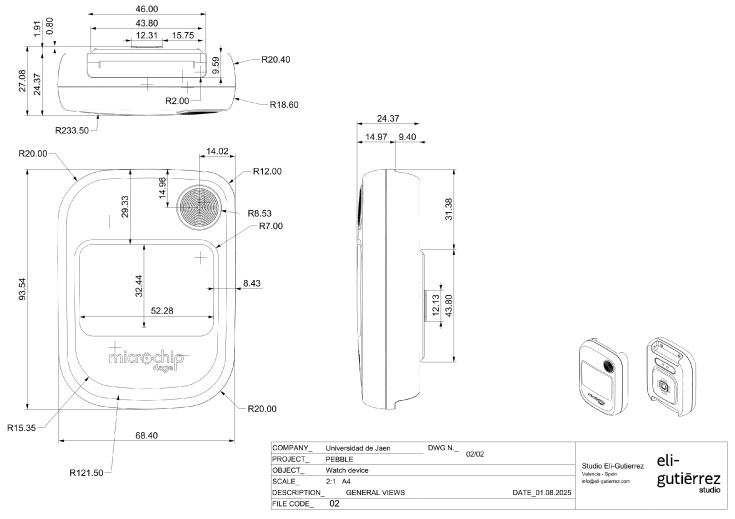
Mechanical design and dimensions of the wristband.

**Figure 4 sensors-25-06499-f004:**
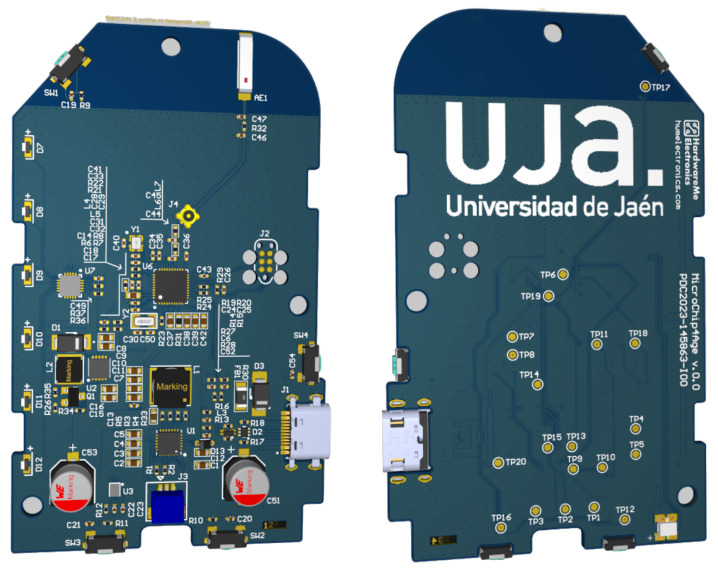
Beacon PCB.

**Figure 5 sensors-25-06499-f005:**

Beacon housing.

**Figure 6 sensors-25-06499-f006:**
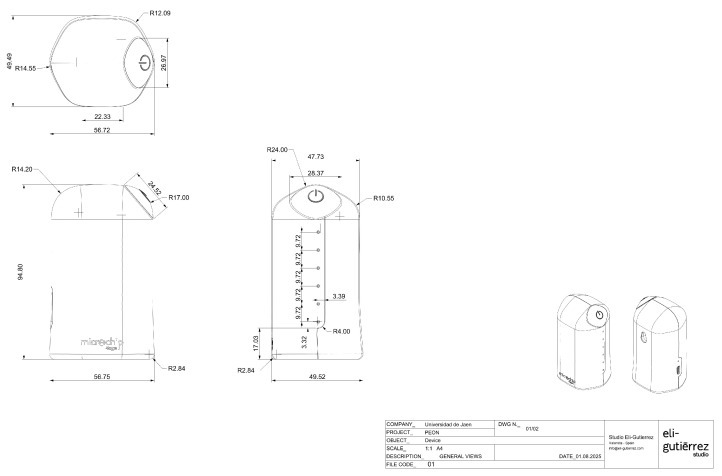
Mechanical layout of the beacon unit.

**Figure 7 sensors-25-06499-f007:**
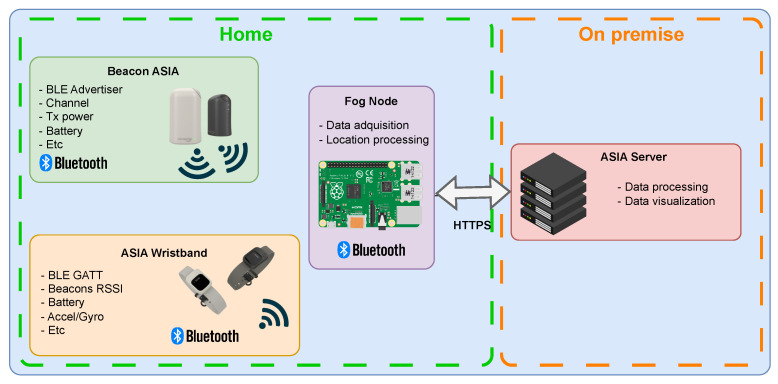
System architecture based on ASIA Beacon and wristband.

**Figure 8 sensors-25-06499-f008:**
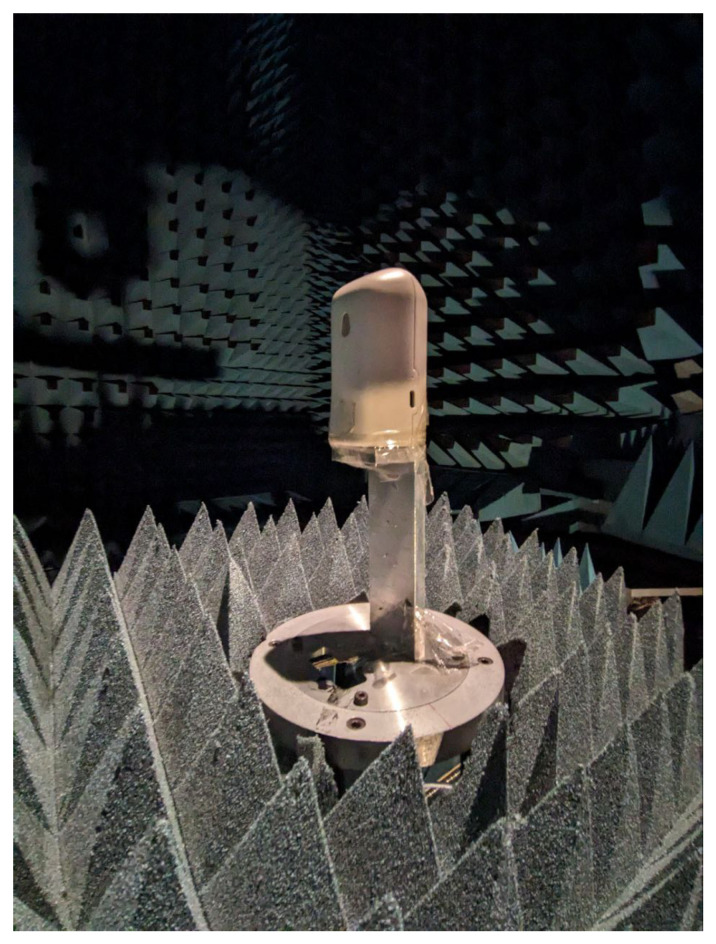
Beacon in anechoic chamber.

**Figure 9 sensors-25-06499-f009:**
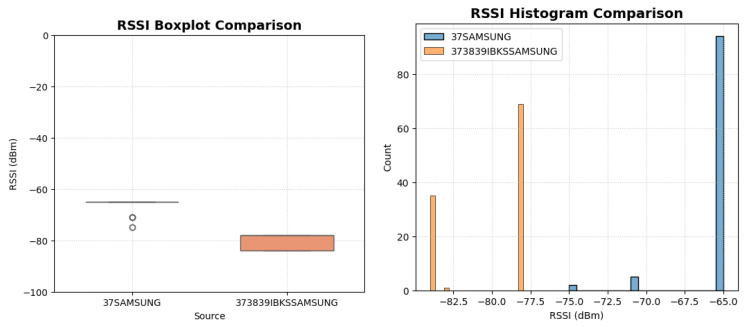
Histogram and Boxplot of RSSI distribution.

**Table 1 sensors-25-06499-t001:** Limitations of commercial BLE beacons and smart wristbands.

Aspect	BLE Beacons	Smart Wristbands
**RSSI instability**	Signal strength varies across channels; affects localization accuracy.	BLE advertising is limited or absent.
**Firmware access**	No support for behavioral updates or reprogramming.	Closed systems with no developer access.
**Energy use**	Optimized for fixed roles, but not adaptable to dynamic passive systems.	High power consumption limits autonomy.
**Sensor data**	Usually do not include sensors or expose data.	Raw sensor access is restricted or unavailable.
**Localization support**	No built-in sync or cooperative positioning features.	Not intended for integration with beacon networks.

**Table 2 sensors-25-06499-t002:** PCB specifications of the flexible section of the wristband.

Parameter	Value
Board dimensions	20.65 mm × 17.65 mm
Number of layers	2 (signal layers; no internal planes)
Total components	19
Components on top side	18
Components on bottom side	1

**Table 3 sensors-25-06499-t003:** PCB specifications of the main section of the wristband.

Parameter	Value
Board dimensions	30.00 mm × 28.50 mm
Number of layers	4 (signal layers; no internal planes)
Total components	131
Components on top side	125
Components on bottom side	6

**Table 4 sensors-25-06499-t004:** Summary of PCB characteristics for the ASIA Beacon prototype.

Parameter	Value
Board dimensions	49.60 mm × 81.90 mm
Number of layers	4 (all signal layers; no internal planes)
Total components	145
Components on top side	124
Components on bottom side	21

**Table 5 sensors-25-06499-t005:** Structure of the Manufacturer Specific Data Payload.

Offset	Size (Bytes)	Type	Name	Description
0–1	2	uint16_t	company_code	Bluetooth SIG-assigned identifier (e.g., 0x0059 for Nordic)
2	1	int8_t	power	Transmit power in dBm (e.g., −20, −8, 0, +4)
3	1	uint8_t	channel	BLE advertising channel used (e.g., 37, 38, 39)
4	1	uint8_t	battery	Battery level as a percentage (0–100)
5–6	2	uint16_t	supply_mv	Measured supply voltage in millivolts
7	1	uint8_t	temperature	Internal temperature in Celsius (integer format)
8	1	uint8_t	flags	Bit-encoded status (e.g., charging, low battery)

**Table 6 sensors-25-06499-t006:** Structure of the BLE Advertising Packet Used in the System.

Field	BLE AD Type	Description
Flags	0x01	General discoverability; BR/EDR mode not supported
Complete Device Name	0x09	Readable device identifier (e.g., BASIA)
Manufacturer Specific Data	0xFF	Application-specific telemetry payload (8–9 bytes)

**Table 7 sensors-25-06499-t007:** Custom GATT Service and Characteristics of the ASIA Wristband.

Characteristic	UUID (16-bit)	Data Type	Unit	Access	Notify
Battery Level	0x2A19	uint8_t	%	Read	Yes
Accelerometer (X, Y, Z)	0xA101	int16[3]	mg	Read	Yes
Heart Rate (BPM)	0x2A37	uint8_t	bpm	Read	Yes
Step Count	0xA102	uint32_t	steps	Read	Yes
Beacon RSSI List	0xA103	array[struct]	dBm	Read	Yes
PPG Raw Data	0xA104	int16_t[n]	ADC counts	Read	Yes

**Table 8 sensors-25-06499-t008:** Software resources used in the system development.

Component	Tool/Framework	Version/Details
Firmware SDK	nRF Connect SDK	v3.1.0
Mobile Application	Flutter SDK	v3.8.0
	Dependencies	flutter_blue_plus 1.35.4
		permission_handler 12.0.0
		fl_chart 1.0.0
		flutter_launcher_icons 0.14.4
		path_provider 2.1.5
		csv 6.0.0
		keep_screen_on 4.0.0
Analysis Software	Python Interpreter	v3.11.9
	pandas	v2.2.3
	numpy	v1.26.4
	scipy	v1.13.1

**Table 9 sensors-25-06499-t009:** Advertising Events and Packet Transmission per Minute.

Interval (ms)	Events/min	Packets/Event	Packets/min	Packets/s
55	1091	3	3273	54.6
105	571	3	1713	28.6
505	119	3	357	5.95

**Table 10 sensors-25-06499-t010:** Captured Packets and Estimated Losses.

Interval (ms)	Captured/min	Captured/s	Losses/min	Capture Rate (%)
55	3109	51.8	164	95%
105	1713	28.6	0	100%
505	357	5.95	0	100%

**Table 11 sensors-25-06499-t011:** Estimated average current consumption for BLE advertising.

Interval (ms)	Duration (ms)	Charge (μC)	Idle (μA)	Average (μA)
50	3.39	10.25	2.0	188
100	3.39	10.25	2.0	100
500	3.39	10.25	2.0	22

**Table 12 sensors-25-06499-t012:** Comparison between ASIA Beacon and iBKS 105.

Aspect	ASIA Beacon (Proposed)	iBKS 105 (Accent Systems)
SoC	Nordic nRF52833 (Cortex-M4F, +8 dBm, BLE 5.1 DF support)	Nordic nRF51822 (Cortex-M0, BLE 4.0)
Firmware	Fully open, Zephyr RTOS; supports OTA/DFU updates	Closed firmware; basic configuration only via vendor app
Advertising control	Channel selection (37/38/39) or combination, adaptive Tx power (−20 to +8 dBm), customizable intervals	Adjustable Tx power and interval (−20 to +4 dBm), customizable intervals
Telemetry	Manufacturer Data includes power, channel, battery, voltage, temperature, flags	Eddystone/iBeacon frames; limited telemetry (battery level)
Power	Lipo 1000 mAh	Coin cell CR2477, 1000 mAh
Battery life	3–4 months continuous operation at 300 ms interval	30–40 months at 1 s interval; 3–4 months at 100 ms interval
OTA updates	Yes (dual-image DFU mechanism)	Limited to vendor firmware releases
Mechanical design	Custom 3D-printed case, Ø94.6 mm × 56.7 mm; wall/surface mounting; LEDs and buttons for local interaction	Plastic enclosure, Ø52.6 mm × 11.3 mm; adhesive mounting; LEDs and buttons for local interaction
Integration	Designed for research and assisted-living systems	Aimed at asset tracking and commercial deployments
Cost	Estimated $50 (components + 3D-printed case, small batch)	Around $19 (retail unit)

**Table 13 sensors-25-06499-t013:** RSSI comparison between ASIA Beacon and iBKS 105.

Device	Mean (dBm)	Std (dB)	Min (dBm)	Max (dBm)
ASIA Beacon (ch. 37)	−65.50	1.88	−75	−65
iBKS 105 (ch. 37–39)	−80.05	2.85	−84	−78

## Data Availability

The datasets presented in this article are not readily available because they are currently part of an ongoing study focused on the analysis and validation of experimental results. Requests to access the datasets should be directed to David Díaz Jiménez (ddjimene@ujaen.es).

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
