# Peer review of "BLE-Based Custom Devices for Indoor Positioning in Ambient Assisted Living Systems: Design and Prototyping"

_sensors, 2025, doi:10.3390/s25206499_

Round 1
Reviewer 1 Report
Comments and Suggestions for Authors
One of the major motivations for the work is to design a customizable BLE-based device in which the multiple parameters can be reprogrammed. However, it is difficult to find this information showing how the proposed design achieves this purpose.
Also, the title of the article is about indoor positioning, but throughout the manuscript the indoor positioning part is less mentioned. How/what did the authors design the indoor positioning method for the designed device?
Section 3 is too verbal to describe the hardware design. Most of the contents are from the used chips. Suggest reducing the size of Section 3 and using a more concise way to describe the hardware, highlighting the new things related to the major motivations of the work, such as adjustable parameters or low-energy consumption.
The experimental results in Section 5 are weak. There is no indoor positioning testing and experiments, which should be the major theme of the work.
The RSSI signals are not stable, though the proposed hardware design optimized them, then how are the authors going to use the RSSI signals for the indoor positioning? Are there any anchor points? How many of them?
Author Response
All answers are included in the attached PDF.

Reviewer 2 Report
Comments and Suggestions for Authors
This study develops a customised BLE device system that offers advantages over commercial solutions in terms of configurability, transparency, and integration. However, it lacks enough experimental details and comparison to verify its effectiveness. Several items need to be further justified or clarified:
- Clarify the novelty compared to existing custom BLE solutions. While the paper clearly outlines the limitations of commercial devices, it would be better to contrast the proposed system with other custom or research-oriented BLE platforms.
- Justify the choice of Zephyr RTOS for your system over other alternatives.
- Line 595:“Up to a 35% reduction in RSSI variability observed when employing selective channel 595 strategies.” is the conclusion. Need more data and analysis.
- Line 597,598: How to “estimate” battery life?
- Table 7: add a column for estimated battery life
- How to quantify the positive impact of the proposed low-energy system on battery life after eliminating the effects of hardware and protocol? What experiments have been conducted to verify the battery life?
- Line 267, you choose nrf 52833, it may not be necessary to explain too much about nrf52832.
- So many repeated definitions. Please check and revise. e.g. “PCB” in line 247 and line 260, “BLE ” in line 40,126,130; “DFU” in line 150,270 (Should the abbreviation be inside or outside the brackets? Please use the correct format.); “RSSI” in line 47 and 134.
- Line 354, why is “six white LEDs” italicised?
- Line 500, why do you explain “GATT” here? You mentioned it many times in the previous content, but did not explain it.
- What is the size of the PCB as shown in Figure 1 and Figure 4?
- The PCB diagram (Figure 1 and Figure 4) does not contain enough information. It is best to briefly explain the functions of different critical modules on the PCB board (such as power, drive, communication, control, etc.) on the design diagram.
Author Response

(The authors gave the same response as above.)

Reviewer 3 Report
Comments and Suggestions for Authors
The manuscript presents a technically rich and relevant contribution to BLE-based indoor localization, with well-designed custom devices and clear potential for Ambient Assisted Living applications. The integration of wearable and beacon systems is thoughtfully executed, and the work demonstrates promising initial results.
- All the Abbreviations should be defined in their first usage only.
- The quality of the Figures (including Figs . 3 and 5) should be improved.
- Strengthen the experimental section with more extensive real-world validation data, particularly focusing on localization accuracy and long-term performance in inhabited smart environments. The current results are limited to controlled laboratory scenarios.
- Include a direct quantitative comparison of the system's localization accuracy against existing state-of-the-art BLE-based indoor localization systems from the literature.
- The work emphasizes advantages but does not critically address possible drawbacks.
- Add a dedicated section discussing the security and privacy measures implemented, particularly in light of the handling of biometric data in Ambient Assisted Living contexts.
- The manuscript reports a 35% reduction in RSSI variability but does not fully describe the method used to achieve this reduction. More detail on the selective channel strategies, antenna design, or filtering methods used is required.
- It is beneficial to benchmark the proposed system against at least one representative commercial device to demonstrate measurable improvements in cost, accuracy, energy consumption, autonomy, and configurability.
- After the introduction section, numerous claims (e.g., regarding the limitations of commercial devices, RSSI variability, or the drawbacks of extended advertising, among others) are made without direct references.
- The conclusion currently restates contributions. It should be expanded to discuss scalability, long-term deployment challenges, and potential improvements more explicitly. Additionally, future directions should be clearly stated.
Comments on the Quality of English Language
- The quality of grammatical and technical writing should be reconsidered and improved.
Author Response

(The authors gave the same response as above.)

Round 2
Reviewer 1 Report
Comments and Suggestions for Authors
- Perhaps the authors can rotate 90 degrees of the Fig. 8 for a better view.
- For the data shown in Table 12, how many RSSI data points were collected to calculate the mean value for each device, respectively? It is difficult to find the information from the context.
- Also, in the response letter, the authors mentioned about the number of anchors used for the indoor positioning experiment, but the reviewer couldn't find this information in the revised manuscript.
- In section 6.5, is it really necessary to mention the professor's name for directing the experiment?
Author Response
The answers are provided in the attached document.

Reviewer 3 Report
Comments and Suggestions for Authors
It is clear that a substantial improvements made in this revised version. The manuscript is clearer, and important additions such as the security/privacy section and benchmarking against a commercial device are valuable. However, some critical issues remain unaddressed correctly:
- The experiments are still limited to laboratory or controlled settings. No long-term evaluation in inhabited smart environments has been provided.
- A direct numerical comparison of localization accuracy with state-of-the-art BLE indoor localization systems is missing. This is essential to establish the contribution relative to existing literature.
- Although some limitations are mentioned, the discussion is still weaker than the emphasis on advantages.
Author Response

(The authors gave the same response as above.)
